# Efficient and Controllable Model Compression through Sequential Knowledge Distillation and Pruning

## Leila Malihi * and Gunther Heidemann

Institute of Cognitive Science, Osnabrück University, 49074 Osnabrück, Germany; gheidema@uos.de
* Correspondence: lemalihi@uos.de

**Abstract:** Efficient model deployment is a key focus in deep learning. This has led to the exploration of methods such as knowledge distillation and network pruning to compress models and increase their performance. In this study, we investigate the potential synergy between knowledge distillation and network pruning to achieve optimal model efficiency and improved generalization. We introduce an innovative framework for model compression that combines knowledge distillation, pruning, and fine-tuning to achieve enhanced compression while providing control over the degree of compactness. Our research is conducted on popular datasets, CIFAR-10 and CIFAR-100, employing diverse model architectures, including ResNet, DenseNet, and EfficientNet. We could calibrate the amount of compression achieved. This allows us to produce models with different degrees of compression while still being just as accurate, or even better. Notably, we demonstrate its efficacy by producing two compressed variants of ResNet 101: ResNet 50 and ResNet 18. Our results reveal intriguing findings. In most cases, the pruned and distilled student models exhibit comparable or superior accuracy to the distilled student models while utilizing significantly fewer parameters.

**Keywords:** knowledge distillation; teacher; student; model compression; parameter; pruning

## 1. Introduction

In the world of deep learning, making models smaller and more efficient is a big challenge [1]. This becomes particularly challenging when deploying DNN models on edge devices such as smartphones and IoT devices. These devices often have limited computing resources and limited memory capacity, which makes it difficult to run complex DNNs efficiently. Model compression techniques, such as knowledge distillation and network pruning, have attracted considerable attention as effective approaches to achieving efficient model deployment while maintaining or even improving model performance [2–4].

Knowledge distillation, introduced by Hinton et al. [1], facilitates knowledge transfer from a large, cumbersome "teacher" model to a smaller and more compact "student" model. The student model learns to mimic the soft labels (probabilities) produced by the teacher, leveraging its knowledge during training to improve performance [2]. Knowledge distillation provides a degree of controllable compression by allowing us to transfer knowledge from a larger model to a smaller one. However, even after knowledge distillation, the model may still retain some redundancy and excess capacity [3]. This is where pruning comes into play. Pruning helps eliminate unnecessary parameters and fine-tune the model further, enabling us to achieve even greater compression without sacrificing performance.

Network pruning, proposed by LeCun et al. [4], selectively removes redundant connections and weights from a neural network, resulting in a more compact and computationally efficient model. While pruning is a powerful technique for model compression, it does not have the ability to accurately control the degree of compression. In other words, you do not have fine-tuned control over how much the model should be compressed. This limitation can lead to trade-offs between compression and performance.

By combining these techniques in a strategic sequence, we can harness the benefits of both knowledge distillation and pruning, ultimately achieving a more efficient and compact

model with controlled compression. Our paper presents a novel method for controlled model compression. We begin by applying knowledge distillation, followed by pruning and fine-tuning. This unique sequence enables us to achieve a more precise and controlled compression of the model while maintaining its performance. By employing pre-trained models from the ImageNet dataset and evaluating them on CIFAR-10 and CIFAR-100, we present a comprehensive analysis encompassing eleven distinct experiments (a notable expansion from the three to four experiments commonly found in existing literature [5–7]). Our primary objective is fourfold:

- We introduce a novel compression framework that begins by distilling knowledge from a large network to a smaller one, followed by pruning and then fine-tuning.
- We introduce the concept of "breakpoint" in our framework, offering a novel perspective on the compression process. Moreover, by employing a wide range of pruning percentages, we ensure compatibility with different scenarios and increase the versatility of our approach.
- The effectiveness of our proposed pipeline is proven through evaluations involving more than 10 model combinations on both CIFAR-10 and CIFAR-100 datasets.
- The results demonstrate the potential to compress the teacher model by over 70% on average, with minor adjustments in accuracy. Notably, some cases even exhibit improved or consistent accuracy.

The motivation behind this research is to address the challenges posed by deploying deep neural networks on edge devices, where computational resources are limited. By optimizing model efficiency through knowledge distillation and network pruning, our research contributes to the growing field of model compression and paves the way for deploying deep learning models on edge devices. The insights gained from this study have implications for various real-world applications where resource-efficient deep learning models are critical.

The rest of the paper is organized as follows. In the Section 2, we provide an overview of the state of the art. Subsequently, we introduce the methodology underlying our experiments in the Section 3, which describes our experimental setup and the approach we adopt. In the Section 4, we present the intricate details of our empirical findings and their broader implications. Finally, we delve into the Sections 5 and 6, where we analyze and summarize our findings, offering additional insights into this task.

## 2. Related Works

Efficient model deployment and model compression have been subjects of extensive research in the deep learning community. Several approaches, including knowledge distillation and network pruning, have been explored to achieve efficient and compact models while preserving performance. In this section, we review relevant studies that have investigated knowledge distillation, network pruning, and a combination of both for model compression.

### 2.1. Knowledge Distillation

The Knowledge Distillation (KD) paper by Hinton et al. [1], introduces the concept of knowledge distillation. It proposes training a smaller model (the student) to mimic the behavior of a larger, more complex model (the teacher) by transferring its knowledge. The teacher model's soft probabilities (logits) are used as "soft targets" during training to guide the student's learning. The approach demonstrates that the student can achieve similar or even superior performance to the teacher despite being much smaller and computationally efficient. Attention Transfer (AT), by Zagoruyko and Komodakis [5], introduces attention transfer for knowledge distillation. Attention maps are used to focus on important regions in the input data. The student model is trained to mimic the attention maps produced by the teacher model. By learning to attend to similar regions, the student can improve their generalization and performance.



Variational Information Distillation (VID), introduced by Ahn et al. [8], incorporates variational distributions in knowledge distillation. The student model learns from the uncertainty present in the teacher's predictions. By considering the variance in the teacher's logits, the student can capture not only the mean representation but also the level of confidence in the teacher's predictions. Contrastive Representation Distillation (CRD), by Tian et al. [2], leverages contrastive learning for knowledge distillation. The teacher and student models are compared using contrastive loss, where positive pairs are from the same class and negative pairs are from different classes. This encourages the student to capture discriminative features while maintaining intra-class similarity and inter-class differences. Similarity-Preserving Knowledge Distillation, by Tung et al. [6], proposes similarity-preserving knowledge distillation, where the student learns to preserve pairwise similarities of the teacher's features during training. By maintaining the relative similarities between the teacher's features, the student model can capture the fine-grained details present in the teacher's representations.

Self-Residual Representation Learning (SRRL), presented by Zhang et al. [6], proposes SRRL for knowledge distillation, focusing on self-residual representations. The teacher's self-residuals are used to enhance the knowledge transfer process. The student learns to capture the teacher's self-residuals, leading to improved generalization. Semantic Conditioned Knowledge Distillation (SemCKD), by Chen et al. [7], introduces semantic conditioning in knowledge distillation. The student model is guided to focus on specific semantic aspects of the teacher's predictions. This enables the student to learn important semantic cues and improve performance on challenging tasks. Simple Knowledge Distillation (SimKD), by Chen et al. [9], proposes utilizing the discriminative classifier directly from the pre-trained teacher model during student inference. Additionally, it involves training the student encoder by aligning its features with the teacher's features using a single $\ell 2$ loss.

Knowledge distillation offers a controllable approach to compressing models by transferring valuable insights from larger models to smaller ones. However, even after knowledge distillation, the model may still contain unnecessary redundancy and excess capacity. This is where pruning comes in, removing unnecessary parameters and tuning the model, enabling us to achieve more compression without losing performance. Through the strategic integration of these techniques, we can leverage the strengths of knowledge distillation and pruning, ultimately creating a more efficient and effective compression model with more precise management.

### 2.2. Pruning

Pruning techniques for deep neural networks have become pivotal for model compression, reducing model size, and enhancing computational efficiency without significant loss in performance. Various methods have been proposed to achieve efficient pruning, and among them, magnitude pruning has gained significant attention. In magnitude pruning, weights with low magnitudes are identified and pruned, leading to sparse networks with reduced parameters. Han and Liu [10] introduce deep compression, incorporating pruning and quantization. Magnitude pruning is employed to remove unimportant weights and achieve model compression.

Molchanov et al. [11], introduce a method to compute the importance of each weight, considering both its magnitude and the layer's sensitivity. The proposed approach improves upon the traditional magnitude-based pruning, providing a more accurate estimate of weight importance and yielding a more compact network. Experimental results demonstrate that the importance estimation approach outperforms other pruning techniques in terms of model compression and efficiency while maintaining comparable predictive performance.

Zhou et al. [12], introduce a comprehensive approach to compressing deep neural networks, combining magnitude pruning, trained quantization, and Huffman coding. The authors propose an iterative pruning technique based on weight magnitudes to obtain a sparse network, followed by fine-tuning for accuracy recovery. Ding et al. [13], present ap-

proximated oracle filter pruning for optimizing the width of convolutional neural networks. Magnitude pruning is used to remove filters with low contributions, enabling destructive width optimization. Frankle et al. [14], introduce the concept of "lottery tickets", a form of global pruning. By pruning entire sub-networks, including neurons and connections, the authors achieve more efficient training and reduce model size. Chen et al. [15], introduce Rocket Launching, a general framework for training lightweight deep neural networks. The authors combine magnitude pruning with learning rate scheduling to optimize network efficiency without sacrificing performance. By iteratively pruning the model and fine-tuning it with a modified learning rate, they achieve a trade-off between model size and accuracy. The proposed approach is shown to be applicable across various network architectures and outperforms existing pruning methods, resulting in more resource-efficient and accurate lightweight models.

While pruning is a powerful tool for model compression, it does not have precise control over the amount of compression. In simpler terms, it does not allow fine-tuning the degree of compression based on specific needs. This often results in a delicate balance between compression and performance. This is where Knowledge Distillation (KD) comes in and offers a solution by introducing a semi-controlled form of compression. Incorporating KD alongside pruning overcomes this limitation and enables a more accurate and balanced compression strategy. Our approach effectively tackles these challenges with a significant advantage. By smartly combining knowledge distillation and pruning, we achieve more compression. Our method leverages the teacher model's distilled knowledge to guide the pruning, eliminating the need for extensive adjustments. Instead, we fine-tune only the last connected layer, resulting in a compact network that outperforms conventional pruning. This demonstrates our method's success in balancing compression and performance.

### 2.3. Combination of Pruning and Knowledge Distillation

Aghli and Ribeiro's work [16] pioneered the integration of weighted pruning and knowledge distillation, coordinating selective pruning on ResNet layers and subsequent distillation for enhanced model compression without loss of accuracy. Xie et al.'s study [17], ventures into the realm of person re-identification. Employing a sequence of pruning followed by knowledge distillation, they strike a balance between effective parameter reduction and accurate performance. Cui and Li, the architects of [18], unveil a complex model compression approach that combines structural pruning with dense knowledge distillation for large language models. Kim et al. [19] address the needs of edge devices with PQK, an innovative combination of pruning, quantization, and knowledge distillation. A structured progression of pruning, quantization, and distillation provides a comprehensive strategy for efficient edge-based model deployment.

Finally, Wang et al. [20] introduce an innovative approach that combines structured pruning with multilevel distillation. By using pre- and post-pruning networks as teacher–student pairs, they reduce the loss of accuracy through distillation and highlight the synergy between the two techniques.

In previous approaches, knowledge distillation is often used for fine-tuning after pruning. This involves using the pruned model as a learner and transferring knowledge from a larger model to modify it. However, this method does not lead to a significant increase in compression. Our approach takes strength from its flexibility and enables a more intelligent design that can increase the compression rate while maintaining or even increasing performance.

However, starting with pruning before distilling knowledge has its drawbacks. This risks losing essential information that distillation can provide, limiting the depth of understanding. In addition, pruning may change the structure of the network, harm teacher–student communication, and affect student learning. On the other hand, starting to distill knowledge before pruning has two clear advantages. First, the learner model takes advantage of advanced knowledge for efficient pruning and improved compression, while preserving important features. Second, distillation provides deep insight and enables finer

pruning that preserves accuracy. This approach shows the potential of using knowledge distillation before pruning and provides a way to achieve advanced model compression with minimal performance downsides.

## 3. Materials and Methods

In this section, we present our new approach designed to achieve efficient model compression while minimizing its impact on accuracy. In Figure 1, we show the proposed model compression pipeline that incorporates our new approach. This pipeline seamlessly integrates knowledge distillation and pruning processes which are followed by fine-tuning to achieve efficient compression while maintaining accuracy.

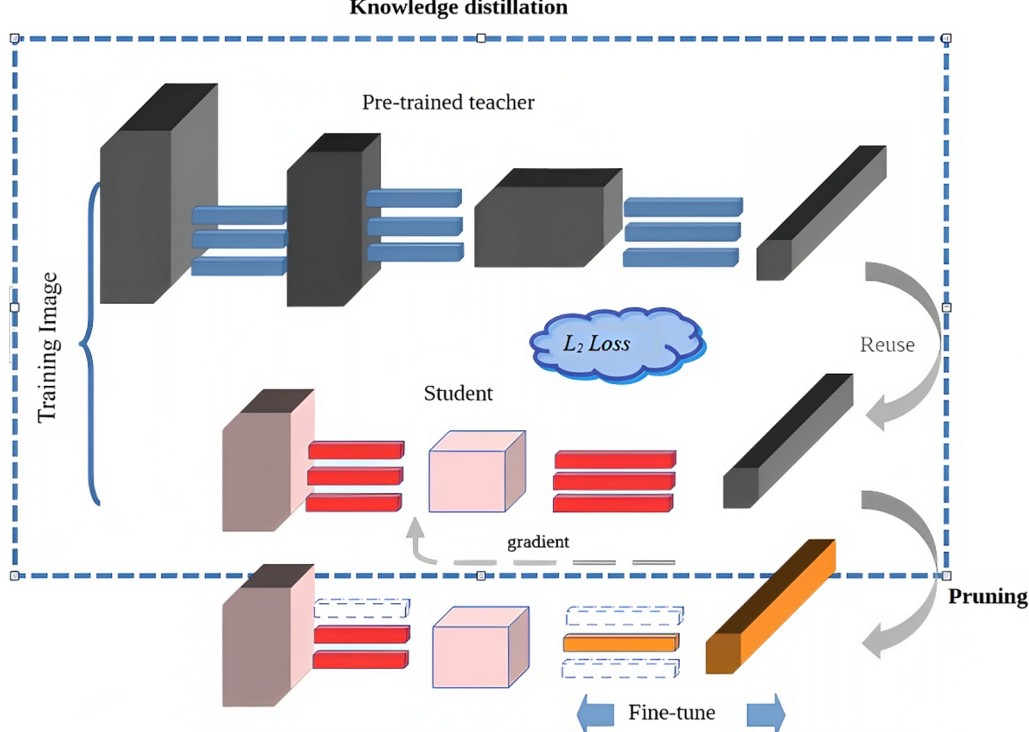

**Figure 1.** Proposed model compression: proposed model compression pipeline with SIMKD Knowledge Distillation, weight pruning, and fine-tuning.

### 3.1. Simple Knowledge Distillation (SimKD)

In recent years, several feature distillation methods have been proposed, aiming to enhance student feature encoders by exchanging additional gradient information between intermediate teacher–student layer pairs. However, these techniques heavily rely on specially crafted knowledge representations to provide suitable inductive bias [7], and meticulous selection of hyper-parameters to balance various losses, making the process time-consuming and labor-intensive.

SimKD [9] introduces a "classifier-reusing" operation, where we directly adopt the pre-trained teacher classifier for student inference, eliminating the need for label information and relying solely on the feature alignment loss to generate gradients and feature extractors are shared across tasks while the final classifier learns task-specific information. SimKD leverages the teacher classifier for inference and enforces feature alignment, enabling effective knowledge transfer to enhance the student model by incorporating the teacher classifier for inference and enforcing their extracted features to align using a specially designed $\ell2$ loss function:

$$L_{simKD} = ||f^t - P(f^s)||^2 \tag{1}$$

A projector was introduced as $P(\cdot)$ [4], which ensures efficient yet accurate feature dimension matching. Surprisingly, this simple technique significantly mitigates the perfor-

mance degradation typically observed in teacher-to-student compression. Using SimKD achieves high inference accuracy, and its single-loss formulation allows for straightforward interpretability. As shown in Figure 1, a fundamental aspect of the proposed method [9], includes the "classifier reuse" operation, where we use the pre-trained teacher classifier for student inference, eliminating the need for training a separate classifier. Importantly, the reused part of the pre-trained teacher model can include more layers beyond just the final classifier.

### 3.2. L1 Unstructured Pruning

Parameter pruning has attracted considerable attention in the field of model compression. In addition to its ability to minimize redundancy, pruning techniques have shown efficacy in reducing overfitting while maintaining versatility in different applications.

L1 unstructured pruning is a technique used to sparsify neural networks by removing individual weights (connections) in a model without considering any specific structural constraints. In this method, the pruning decision is solely based on the magnitude of each weight, measured using the L1 norm. The L1 norm of a weight vector is the sum of the absolute values of its elements [12]. The formula for the L1 norm of a weight vector w of length n is given by:

$$\text{L1 norm (w)} = |w_1| + |w_2| + ... + |w_n| \tag{2}$$

L1 unstructured pruning involves pruning a specific percentage of weights with the smallest L1 norm values, effectively reducing them to zero. This results in a sparse model where many weights become non-trainable (zero-valued) and are not considered during model inference. Mathematically, the pruning process can be expressed as follows:

1.  Sort the weights in the neural network in ascending order based on their L1 norm values.
2.  Determine the pruning threshold by selecting the L1 norm value corresponding to the desired pruning percentage (e.g., 50% pruning means removing half of the smallest magnitude weights).
3.  Set all the weights with L1 norm values below the pruning threshold to zero.

After the pruning process, the sparse model with zero-valued weights can be fine-tuned or used directly for inference. L1 unstructured pruning offers several advantages, such as reducing model size, improving computational efficiency, and potentially providing opportunities for hardware acceleration due to the sparsity. As depicted in Figure 1, once the knowledge distillation process is complete, we proceed with L1-unstructured pruning.

### 3.3. Fine-Tune

The final step of our method is to apply fine-tuning to the last fully connected layer. Given that our models are initially pre-trained on the ImageNet dataset, the next fine-tuning step is performed specifically on the CIFAR-10 and CIFAR-100 datasets. Our goal was to evaluate the performance of the model using a new dataset, in this way, the strength and effectiveness of our method are evaluated on a different dataset. This step was necessary to demonstrate the adaptation of the method beyond its primary educational context.

Fine-tuning of the last fully connected layer offers two distinct advantages. First, it allows the model to adapt and optimize its features to the specific features of the new dataset, thereby increasing its performance in CIFAR-10 and CIFAR-100. Second, unlike many pruning methods that involve retraining all layers to achieve the desired accuracy [1,4], our approach avoids fine-tuning the entire layer. This not only saves computational resources and valuable time but also preserves the benefits gained from initial pre-training on ImageNet and contributes to the efficiency of our compression pipeline.

## 4. Experiments

### 4.1. Datasets and Baselines

For our experiments, we use two benchmark image classification datasets: CIFAR-100 [21] and CIFAR-10 [21]. To ensure consistency, we apply standard data augmentation techniques and normalize all images using channel means and standard deviations, as commonly done in prior works [17,18]. We compare various knowledge distillation approaches with the vanilla knowledge distillation (KD) [1] baseline, Attention Transfer (AT) [5], Similarity-Preserving Knowledge Distillation (SP) [3], Variational Information Distillation (VID) [8], Contrastive Representation Distillation (CRD) [7], Self-Regularized Knowledge Distillation (SRRL) [6], Semantic Consistency Knowledge Distillation (SemCKD) [7], and Simple Knowledge Distillation (SimKD) [9].

### 4.2. Training Details

We use the SGD optimizer with 0.9 Nesterov momentum for all datasets based on the [9]. We set the total training epoch to 50 and adjust the learning rate to 0.001. The mini-batch size is set to 64, and the weight decay is set to $5 \times 10^{-4}$. The temperature T in the KD loss is consistently set to 4 based on the [9]. All the experiments are done in Pytorch 2.0.0, CUDA 12.0, and CUDNN 8.9.3, with Quadro RTX 8000 GPU.

### 4.3. Distillation

Tables 1 and 2 provide a comprehensive performance comparison of various distillation approaches using 11 network combinations where teacher and student models have either similar or completely different architectures. We compare nine different knowledge distillation methods to determine which one gives the best performance. We observe that SimKD consistently outperforms all competitors on CIFAR-100 and CIFAR-10. Then, we choose these distilled students for the next step and apply pruning to that.

**Table 1.** Distil knowledge with different methods on the CIFAR-100.

| Student | Res [2] 50 | Res 18 | Res 34 | MobileNet | MobileNet | Dens 169 | Dens 201 | Effici [1] b0 | Effici b2 | Effici b4 | MobileNet |
|---|---|---|---|---|---|---|---|---|---|---|---|
| KD [1] | 58.32 | 54.13 | 50.35 | 51.53 | 52.25 | 54.32 | 52.78 | 56.3 | 53.78 | 58.56 | 51.42 |
| AT [5] | 58.45 | 54.32 | 50.13 | 51.21 | 52.17 | 54.56 | 53.4 | 56.54 | 53.80 | 58.34 | 51.12 |
| similarity [3] | 58.12 | 54.47 | 50.34 | 51.46 | 52.56 | 54.78 | 53.1 | 56.23 | 54.23 | 58.67 | 51.38 |
| VID [8] | 58.67 | 54.23 | 50.43 | 51.12 | 52.67 | 55.1 | 53.56 | 56.21 | 54.45 | 58.58 | 51.56 |
| CRD [2] | 58.69 | 54.56 | 50.49 | 51.59 | 52.48 | 55.32 | 53.47 | 56.65 | 54.39 | 58.76 | 51.18 |
| SRRL [6] | 58.79 | 54.78 | 50.67 | 51.76 | 52.67 | 55.23 | 53.76 | 56.73 | 54.74 | 58.85 | 51.69 |
| SemCKD [7] | 58.87 | 54.83 | 50.91 | 51.65 | 52.78 | 55.67 | 53.87 | 56.72 | 54.71 | 58.72 | 51.87 |
| SimKD [9] | **59.12** | **55.6** | **51.3** | **52.1** | **53.2** | **56.45** | **54.78** | **57.1** | **55.87** | **59.4** | **52.07** [3] |
| Teacher | Res101 54.07 | Res101 54.07 | Res152 52.24 | Dens 169 57.2 | Dense121 57.1 | Dens161 58.2 | Dens161 58.2 | Effici b1 59.2 | Effici b3 58.1 | Effici b5 59.5 | Effici b3 58.1 |

[1] Effici: EfficientNet. [2] Res: ResNet. [3] The bold one is the best result.

**Table 2.** Distil knowledge with different methods on CIFAR-10.

| Student | Res 50 | Res 18 | Res 34 | MobileNet | MobileNet | Dens 169 | Dens 201 | Effici b0 | Effici b2 | Effici b4 | MobileNet |
|---|---|---|---|---|---|---|---|---|---|---|---|
| KD [1] | 80.14 | 79.36 | 79.54 | 75.93 | 78.59 | 81.69 | 81.54 | 78.81 | 80.08 | 82.03 | 78.17 |
| AT [5] | 80.45 | 79.12 | 79.32 | 75.89 | 78.87 | 81.98 | 81.23 | 78.71 | 80.41 | 82.16 | 78.41 |
| similarity [3] | 80.32 | 79.56 | 79.56 | 76.12 | 79.54 | 82.24 | 81.78 | 79.25 | 80.32 | 82.43 | 79.36 |
| VID [8] | 80.67 | 79.67 | 79.73 | 76.32 | 79.32 | 82. 36 | 81.85 | 79.47 | 80.58 | 82.51 | 79.59 |
| CRD [2] | 80.78 | 79.78 | 79.68 | 76.67 | 79.71 | 82.41 | 81.79 | 79.61 | 80.69 | 82.68 | 79.51 |
| SRRL [6] | 80.69 | 79.81 | 79.83 | 76.86 | 79.79 | 82.67 | 81.91 | 79.69 | 80.71 | 82.72 | 79.54 |
| SemCKD [7] | 80.89 | 79.92 | 79.74 | 77.1 | 79.82 | 82.54 | 81.81 | 79.72 | 80.79 | 82.79 | 79.61 |
| SimKD [9] | **81.7** | **80.56** | **80.1** | **78.2** | **80.21** | **82.76** | **82.33** | **80.2** | **81.14** | **83.3** | **80.2** |
| Teacher | Res101 81.1 | Res101 81.1 | Res152 81.42 | Dens169 83.15 | Dense121 83.11 | Dens 161 83.07 | Dens 161 83.07 | Effici b1 81.1 | Efficib3 80.5 | Effici b5 83.4 | Effici b3 80.5 |

*4.4. Pruning*

In our investigation, we conducted network pruning experiments on the distilled student model using the SimKD approach [9]. The pruning process involved gradually reducing the number of parameters in the model from 0% to 90% in 10% increments. This systematic approach allowed us to explore the impact of varying levels of parameter reduction on the model's performance and efficiency. We carried out pruning 10 times for each teacher–student pair and then took the average result to improve the reliability of our analysis, reduce the dependence on the Initial Model Capacity, reduce the impact of random initialization, and ensure a more representative evaluation of the knowledge transfer process. It allows us to obtain a more stable estimate of the model's performance after pruning, which is especially important if the pruning process is sensitive to the starting configuration of the model.

After pruning, calculate and record the number of parameters remaining in the pruned teacher and student network. This can be done by counting the non-zero weights or connections in the network. Figures 2 and 3, show the plot of accuracy against the remaining parameters, visualizing the trade-off between model size and accuracy. As the model is pruned or compressed, its parameter count decreases, and ideally, the accuracy should decrease gradually or remain relatively stable. However, at a certain point in the plot, a breakpoint may occur.

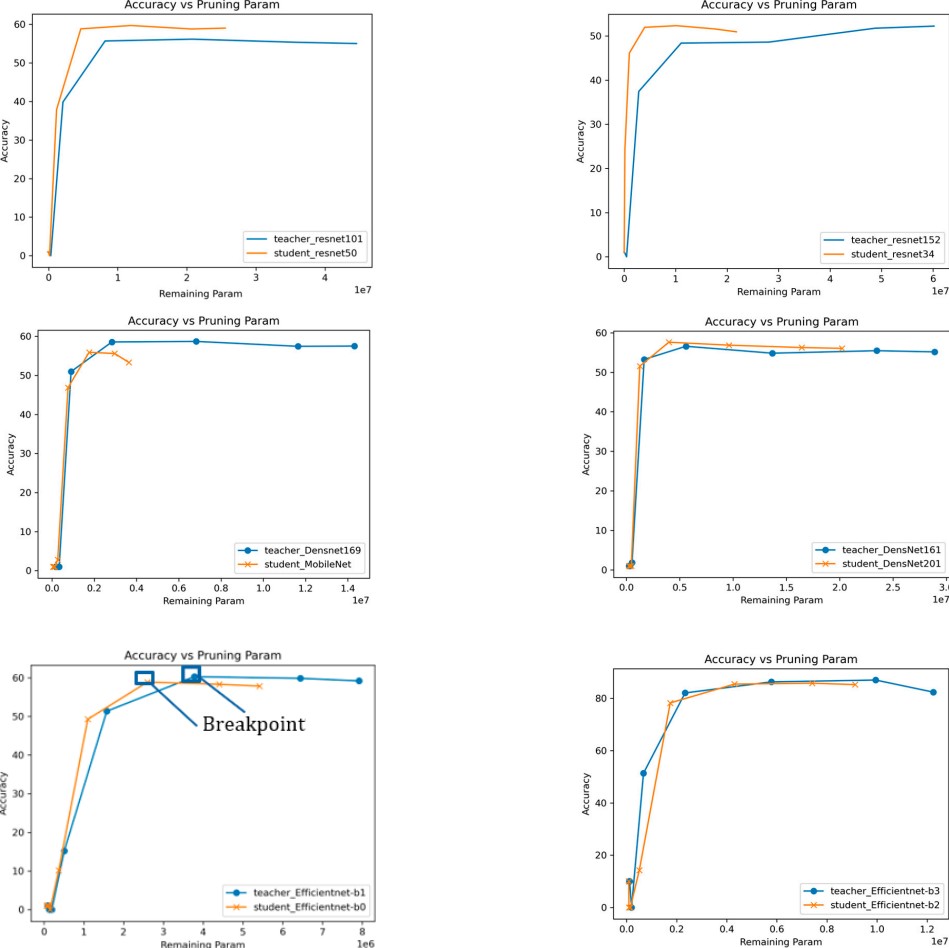

**Figure 2.** Pruning analysis: a comparative study of remaining parameters in various student and teacher models on the CIFAR-100 dataset.

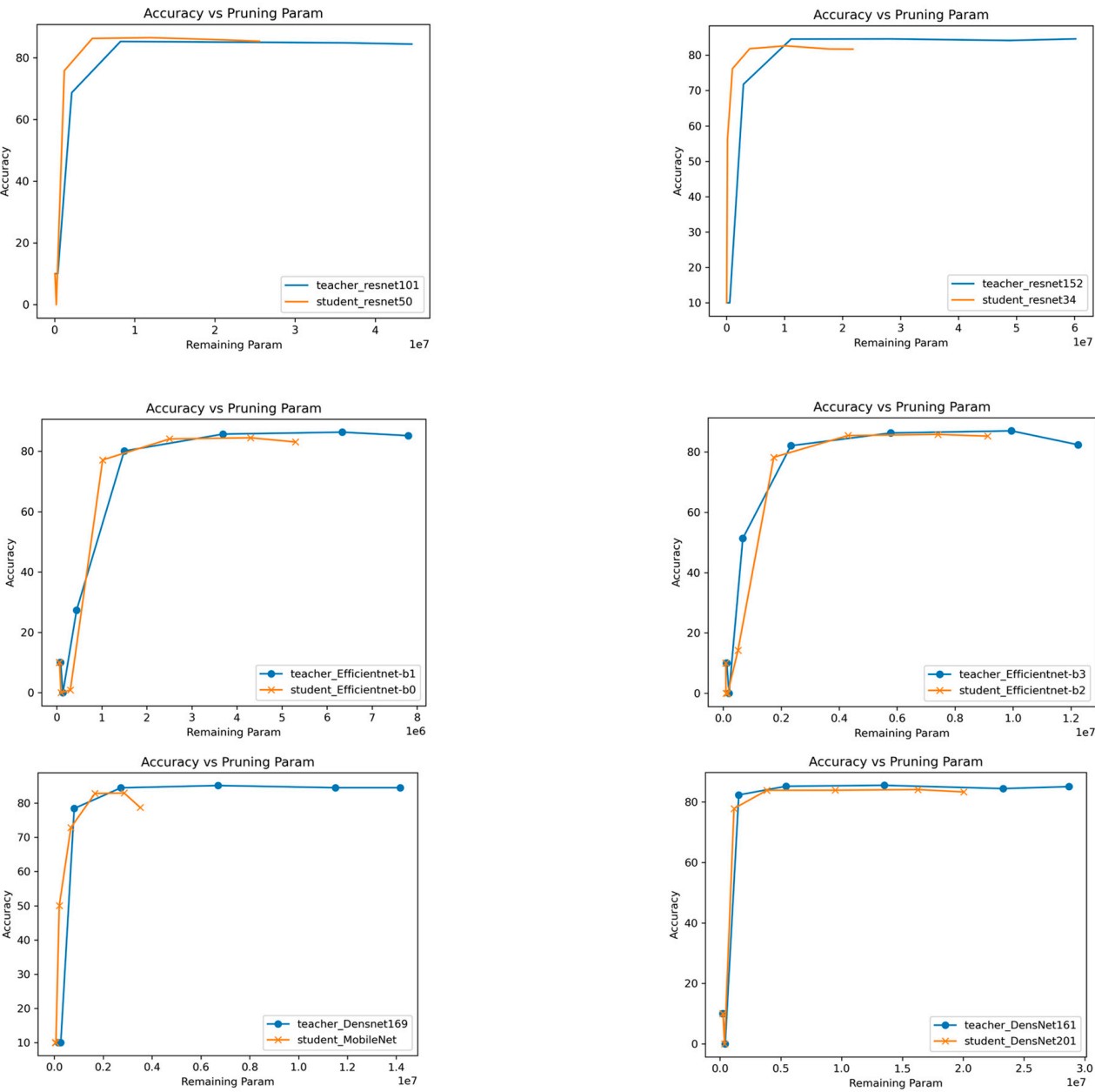

**Figure 3.** Pruning analysis: a comparative study of remaining parameters in various student and teacher models on the CIFAR-10 dataset.

The **Breakpoint** represents a critical threshold where further compression leads to a more rapid decrease in accuracy relative to the reduction in model size (parameters). As the compression rate increases and more parameters are pruned, a point is reached where further reductions in model size may lead to a disproportionate decrease in accuracy. This threshold, known as the breakpoint, serves as an important guide to optimizing the trade-off between model compactness and performance. This helps us identify the sweet spot where compression benefits are maximized while maintaining acceptable levels of accuracy.

In other words, beyond this breakpoint, the decrease in accuracy becomes more significant for a relatively smaller reduction in model size. In the presented figures (Figures 2 and 3), we observe intriguing trends in the accuracy-performance trade-off of the student models. In many cases, the student curve consistently remains above the teacher curve, and the breakpoint of the student curve consistently occurs before the break-

point of the teacher curve. This observation indicates significant compression without sacrificing accuracy.

Tables 3 and 4 offer a comprehensive summary of our experimental outcomes on the CIFAR-10 and CIFAR-100 datasets, showcasing different model architectures. These experiments encompassed the processes of knowledge distillation and network pruning. The tables provide a clear comparison of accuracy and parameter counts for both teacher and student models, both before pruning and at the breakpoint following pruning. When analyzing the results, we observe a consistent trend across both datasets: the pruned student models often achieve the same or higher accuracy than the teacher models, while using fewer parameters.

**Table 3.** Accuracy and number of parameters for different models on CIFAR-100.

| Teacher, Student | Res101, Res50 | Res152, Res 34 | Res101, Res 18 | Dens169, Mobile | Dens121, Mobile | Dens161, Dens 169 | Dens161, Dens 201 | Effici b1, Effici b0 | Effici b3, Effici b2 | Effici b3, Mobile [2] | Effici b5, Effici b4 |
|---|---|---|---|---|---|---|---|---|---|---|---|
| ACC(t) ACC(s)$_{distillation}$ | 54.07 59.12 | 52.2 51.3 | 54.07 55.6 | 57.2 52.1 | 57.1 53.2 | 58.2 56.45 | 58.2 54.78 | 59.2 57.1 | 58.1 ± 0.5 55.87 | 58.1 52.07 | 59.5 59.4 |
| P [1] (t)$_{before pruning}$ × $10^7$ P(s)$_{before pruning}$ × $10^7$ | 4.45 2.56 | 6.02 2.13 | 4.45 1.17 | 1.42 0.42 | 0.8 0.42 | 2.68 1.42 | 2.68 2.0 | 0.78 0.56 | 1.22 0.91 | 1.22 0.42 | 3.04 1.93 |
| ACC(t)$_{breakpoint}$ ACC(s)$_{breakpoint}$ | 54.34 58.92 | 47.48 53.47 | 53.15 55.25 | 57.95 55.1 | 57.8 55.6 | 57.71 57.61 | 58.94 55.23 | 60.17 58.63 | 60 55.91 | 60 54.8 | 60.06 59.8 |
| P(t)$_{breakpoint}$ × $10^7$ P(s)$_{breakpoint}$ × $10^7$ | 0.91 0.5 | 1.1 **0.37** | 0.91 **0.32** | 0.35 **0.18** | 0.18 **0.19** | 0.5 **0.29** | 0.52 0.41 | 0.39 **0.27** [4] | 0.6 0.42 | 0.55 **0.15** | 1.42 **0.8** |
| Percentage Decrease [3] (%) | 88.76 | 93.85 | 92.84 | 87.32 | 76.25 | 89.19 | 84.7 | 65.38 | 67.23 | 87.70 | 73.68 |
| Compression rate | 8.9× | 5.7× | 13.9× | 7.8× | 4.2× | 9.2× | 6.5× | 2.8× | 2.9× | 8.1× | 3.8× |

[1]. P: number of parameters P(t): number of parameters in the teacher; P(s): number of parameters in the student. [2]. Mobile: MobileNet. [3]. Percentage Decrease: the amount of decreased parameters from teacher to student = ((P(t)$_{before pruning}$ − P(s)$_{breakpoint}$)/P(t)$_{before pruning}$) × 100. [4]. The bold one presents the most compressed version of the teacher.

**Table 4.** Accuracy and number of parameters for different models on CIFAR-10.

| Teacher, Student | Res101, Res50 | Res152, Res 34 | Res101, Res 18 | Dens169, Mobile | Dens121, Mobile | Dens161, Dens 169 | Dens161, Dens 201 | Effici b1, Effici b0 | Effici b3, Effici b2 | Effici b3, Mobile | Effici b5, Effici b4 |
|---|---|---|---|---|---|---|---|---|---|---|---|
| ACC(t) ACC(s)$_{distillation}$ | 81.1 81.7 | 81.42 80.1 | 81.1 80.65 | 83.15 78.2 | 83.11 80.21 | 83.07 82.76 | 83.07 82.33 | 81.1 80.2 | 80.5 81.14 | 80.5 80.2 | 83.4 83.3 |
| P(t)$_{before pruning}$ × $10^7$ P(s)$_{before pruning}$ × $10^7$ | 4.45 2.56 | 6.02 2.13 | 4.45 1.17 | 1.42 0.42 | 0.8 0.42 | 2.68 1.42 | 2.68 2.0 | 0.78 0.56 | 1.22 0.91 | 1.22 0.42 | 3.04 1.93 |
| ACC(t)$_{breakpoint}$ ACC(s)$_{breakpoint}$ | 82.7 83.51 | 83.15 80.71 | 83.32 80.78 | 83.53 82.93 | 82.87 80.72 | 82.86 83.02 | 83.53 83.02 | 81.62 80.74 | 80.52 81.15 | 80.12 80.13 | 83.1 83.78 |
| P(t)$_{after pruning}$ × $10^7$ P(s)$_{after pruning}$ × $10^7$ | 0.91 0.6 | 1.1 **0.51** | 0.91 **0.34** | 0.25 **0.16** | 0.16 **0.16** | 0.38 **0.2** | 0.58 0.41 | 0.37 **0.27** | 0.47 0.27 | 0.22 **0.18** | 1.4 **0.8** |
| Percentage Decrease (%) | 86.51 | 91.5 | 92.35 | 88 | 80 | 92.5 | 84.7 | 65.3 | 77.86 | 85.2 | 73.68 |
| Compression rate | 7.41× | 11.76× | 13.08× | 8.87× | 4.9× | 13.4× | 6.7× | 2.8× | 4.5× | 6.7× | 3.8× |

This trend is particularly obvious in cases like ResNet101 vs. ResNet50 and ResNet152 vs. ResNet34 in CIFAR-100, where the pruned student models outperform their respective teacher models in terms of accuracy with a notable reduction in parameters. In the CIFAR-100 experiments, the pruned student model of ResNet101 (ResNet50) achieved an accuracy of 58.92% with just $0.5 \times 10^7$ parameters. In contrast, the teacher model had an accuracy of 54.07% and $4.45 \times 10^7$ parameters. This means a significant reduction of 88.76% in the number of teacher parameters. Similarly, in the CIFAR-10 experiments, the pruned student model of ResNet101 (ResNet50) achieved an accuracy of 83.51% with only $0.6 \times 10^7$ parameters. In comparison, the teacher model had an accuracy of 81.1% and $4.45 \times 10^7$ parameters. This signifies a substantial reduction of 86.51% in the number of teacher parameters.

In addition to the consistent trend mentioned earlier, we observed some specific cases in the CIFAR-100 and CIFAR-10 experiments where the pruned and distilled student models demonstrated superior performance compared to the distilled student models, even with fewer parameters. In CIFAR-100, one such case is with the Densenet

161 architecture. The distilled student model (Densenet169) achieved an accuracy of 56.45% with $1.42 \times 10^7$ parameters, whereas the pruned and distilled student had an accuracy of 57.61% with a 4.8 times smaller number of parameters ($0.29 \times 10^7$). This indicates that the pruned student model not only had better efficiency but also better accuracy than the distilled student model.

For Densenet169, the pruned student (Mobilenet) model achieved an accuracy of 82.93% with $0.16 \times 10^7$ parameters, while the teacher model obtained an accuracy of only 83.15% with the $1.42 \times 10^7$ number of parameters, which means the number of parameters in teacher decreased by 88%.

Table 5 presents a comprehensive comparison of the parameters for all teacher models after pruning. For instance, in the case of ResNet101, our compact version utilizes $0.32 \times 10^7$ parameters, whereas in the previous work [14], this count is $1.78 \times 10^7$, and in another study [22], it is $1.73 \times 10^7$. It is evident that our method consistently produces a more compact version of ResNet101. The consistent findings highlight the effectiveness of combining knowledge distillation and network pruning for making models more efficient without sacrificing performance. Knowledge distillation helps student models learn valuable insights from teachers, which proves particularly useful during pruning.

**Table 5.** Comparison number of parameters.

| Teacher | Res101 | Res152 | Dens169 | Dens121 | Dens161 | Effici b1 | Effici b3 | Effici b5 |
|---|---|---|---|---|---|---|---|---|
| **CIFAR-100(ours)** $\times 10^7$ | **0.32** [1] | 0.37 | 0.18 | 0.19 | 0.29 | 0.27 | 0.15 | 0.8 |
| **CIFAR-10(ours)** $\times 10^7$ | 0.34 | 0.51 | 0.16 | 0.16 | 0.2 | 0.27 | 0.18 | 0.8 |
| [14] $\times 10^7$ | 1.78 | - | - | - | - | - | - | - |
| [22] $\times 10^7$ | 1.73 | - | - | - | - | - | - | - |

[1]. That is the best result.

## 5. Discussion

Our approach provides a clear advantage by enabling a greater compression of the teacher model compared to the previous methods that relied on pruning [10–15], knowledge distillation (KD) [1,3,5,7,8], and the combination of both [16–20]. The crucial aspect of our study is pruning followed by KD. This sequence optimizes the compression efficiency through a two-step process. First, we can choose a smaller network via KD and then further improve the compression by pruning. In contrast, methods that apply KD after pruning generally employ a single compression step.

In this scenario, pruning is usually performed to simplify the model, followed by KD primarily for fine-tuning purposes. Our method is distinguished by its distinct two-step compression process. Our method successfully reduces the number of parameters in models while maintaining their accuracy. For example, in experiments with ResNet architectures, we saw a significant reduction in parameter count. In one case, the student model, which is a pruned version of ResNet18, had over 92.5% fewer parameters compared to its teacher model, ResNet101. This trend demonstrates the effectiveness of our approach in creating more compact models.

Our results are also shown visually in Figures 4 and 5. These figures provide a comparison between the accuracy of teacher and student models, besides the reduction in the number of parameters from teacher to student (Percentage Decrease). These figures demonstrate a consistent pattern across all models.

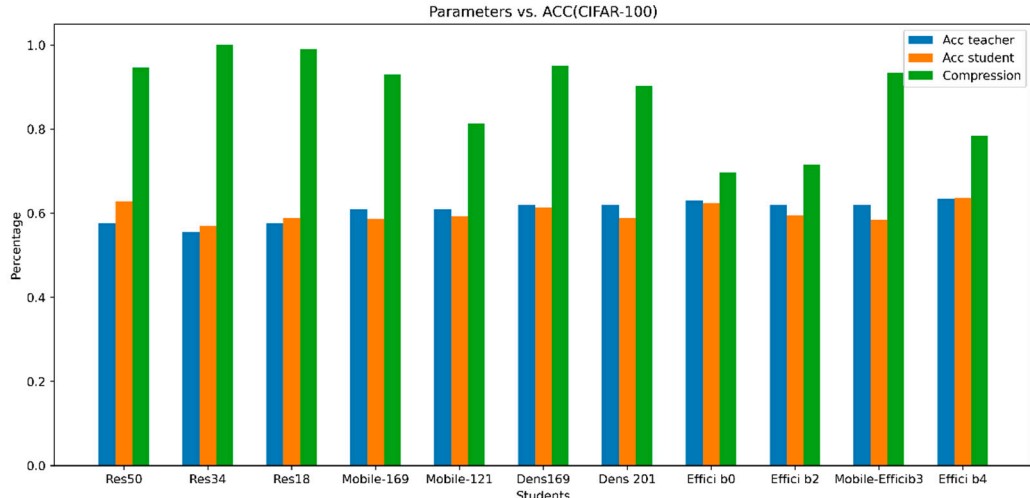

**Figure 4.** Comparison of teacher and student accuracy and reduction in the number of parameters on CIFAR-100.

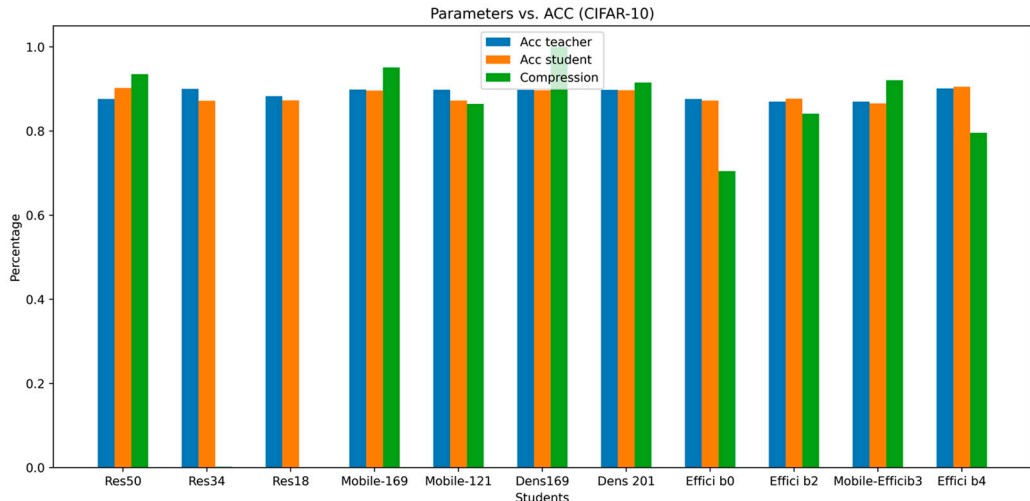

**Figure 5.** Comparison of teacher and student accuracy and reduction in the number of parameters on CIFAR-10.

The accuracy of the student models is very close to or even exceeds the accuracy of the teacher models while achieving a significant reduction in the number of parameters, often by more than 65%, and sometimes even over 90%.

We also found that we can create different compressed versions of the same teacher model. For instance, with DenseNet161, we can make two compressed versions: one from DenseNet169 and another from DenseNet201. Both achieve the same accuracy, but the first one is considerably more compact, approximately half the size of DenseNet201. This flexibility lets us choose the right balance between model size and accuracy based on our needs.

It is worth mentioning that in our results, we have two versions of the compressed ResNet101 model. One has a 7.41 compression rate, pruned from ResNet50. The other has a 13.08 compression rate, pruned from ResNet18. While the pruned ResNet50 model is very accurate at 83.5%, outperforming both ResNet101 and ResNet18, the pruned ResNet18 is also quite accurate at 80.78%, just 1% lower than its original version. This indicates that when selecting a compressed model for ResNet101, the choice depends on our priorities. If we are looking for a more compact model of ResNet101 with just a 1% decrease in accuracy compared to the original, then ResNet18 is the ideal choice. However, if we value accuracy more than compactness, then ResNet50 is the better option.

Furthermore, in our method fine-tuning the last fully connected layer offers two significant advantages. First, it allows the model to fine-tune its properties according to the specific characteristics of new datasets such as CIFAR-10 and CIFAR-100, resulting in improved performance. Second, unlike many pruning methods that require retraining all layers to achieve the desired accuracy [23], our approach does not involve fine-tuning the entire network. This not only saves computational resources and time but also preserves the benefits of initial training on ImageNet and increases the efficiency of our compression process.

## 6. Conclusions

In this study, we explored the advantages of employing knowledge distillation before pruning, resulting in more compact models with controllable levels of compactness. We successfully created two compact versions of ResNet101 using ResNet18 and ResNet50, showcasing the flexibility of our approach. Our unique compression framework follows a sequential process: knowledge distillation, pruning of the distilled student model, and fine-tuning. The results highlighted the potential to achieve more than 70% compression of the teacher model across 11 different model combinations. Intriguingly, some cases even showed improved or consistent accuracy.

## 7. Future Research

For future research directions, we recommend considering the combination of Knowledge Distillation (KD) with other advanced pruning techniques to explore their collective impact on model performance. This could involve integrating KD with methods such as Structured Pruning, which targets specific patterns within the network, or Channel Pruning, which focuses on removing entire channels. Investigating the synergistic effects of these techniques might yield even more efficient models with enhanced accuracy. Additionally, delving into the optimization of the fine-tuning process after pruning could be valuable. Exploring different fine-tuning strategies, such as adjusting learning rates, training epochs, or even considering adaptive learning rates for different layers, could potentially lead to better model convergence and overall performance. Furthermore, extending the evaluation to a wider range of datasets, including more diverse and challenging ones, would provide a deeper understanding of the generalizability of the proposed approach. Assessing the method's effectiveness across various domains could reveal insights into its adaptability and robustness.

**Author Contributions:** Conceptualization, L.M.; methodology, L.M.; software, L.M.; validation, L.M.; formal analysis, L.M.; investigation, L.M.; resources, L.M. and G.H.; data curation, L.M.; writing—original draft preparation, L.M.; writing—review and editing, L.M. and G.H.; visualization, L.M.; supervision, G.H.; funding acquisition, L.M. All authors have read and agreed to the published version of the manuscript.

**Funding:** This paper was funded by Osnabrück University, Open Access Program.

**Data Availability Statement:** Researchers interested in accessing the data may contact the corresponding author and the institutional review board for potential collaboration and data access approval.

**Conflicts of Interest:** The authors state that there are no conflict of interest related to the study. The funders played no role in the study's design, data collection, analysis, interpretation, manuscript writing, or decision to publish. The study was solely supported by the authors' institutions. This declaration ensures transparency and independence in the research process.

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
