# Peer review of "Efficient and Controllable Model Compression through Sequential Knowledge Distillation and Pruning"

_2504-2289, doi:10.3390/bdcc7030154_

Round 1

Reviewer 1 Report

The above article propose an optimizing model efficiency by combining knowledge Distillation and Network pruning for improved generalization. The paper is well research although considerations must be taken into account. 

Major issues

Overall, it is difficult to evaluate the novel contribution of the paper. Have the authors solely compared the performance of different student and teacher models? The authors must emphasize in the paper their contribution and the novelty.

The primary objectives were the evaluation of the combined impact of knowledge distillation and network pruning on model generalization and assess the efficiency gains achieved through reduced model parameters but failed in describing the model generalizations.

Some paragraphs are repetitive and must be rewritten. See for instance: “Knowledge Distillation (KD) paper by Hinton et al. [1] introduces the concept of knowledge distillation”.

Tables 4.1 and 4.2. Please, detail what the numbers in the table represent.

Conclusions: The authors stated that “In the majority of cases, the pruned and distilled student models achieved comparable or even superior accuracy to the distilled student models, despite having significantly fewer parameters. These findings challenge conventional assumptions about model capacity, highlighting the potential of knowledge distillation and pruning high-performing, resource-efficient models for deployment on edge devices with limited computational resources”

However, the authors didn’t compare or contrast the memory requirements, or the computational resources needed for each model.

Author Response

Hello,

I changed most of the part you mentioned, the table 1 and 2 represent:  

performance comparison of various distillation approaches using 11 network combinations where teacher and student models have either similar or completely different architectures. We compare 9 different knowledge distillation methods to determine which one gives the best performance. I will upload the new version of the paper.

Reviewer 2 Report

In this paper, the authors explored the combination of knowledge distillation and network pruning for achieving efficient model deployment in deep learning. But supplementation and additional explanations seem necessary for the following concerns.

1.In Section 4, you mentioned that, the learning rate was set to 0.001, the mini-batch size to 128, and the weight decay to 5 × 10-4. How did you select these hyperparameters and did you do any offline hyperparameter tuning? It may be helpful to mention this at this point in the text. 

2.The article seems to lack some innovation. It is suggested that authors should clearly mention the literature gaps and novelty of the proposed work. 

3.The references of the article are few and not new enough.

4.Future research should be included at the end of conclusion. 

5.In order to better support the author's conclusion, it is recommended to introduce the experiment simulation environment.

The organizational skills and writing level of the article need to be improved.

Author Response

1.In Section 4, you mentioned that, the learning rate was set to 0.001, the mini-batch size to 128, and the weight decay to 5 × 10-4. How did you select these hyperparameters and did you do any offline hyperparameter tuning? It may be helpful to mention this at this point in the text. 

* I mentioned in the text

2.The article seems to lack some innovation. It is suggested that authors should clearly mention the literature gaps and novelty of the proposed work. 

* I changed the introduction and related work and added more references

3.The references of the article are few and not new enough.

4.Future research should be included at the end of conclusion. 

* I added

5.In order to better support the author's conclusion, it is recommended to introduce the experiment simulation environment.

* I did

Reviewer 3 Report

1. The maunscript provides an interesting approach to model compression and is highly relevant to existing state-of-the-art in synergy-based approaches without compromising on accuracy.

2. Synergy between knowledge distillation and network pruning to achieve optimal model efficiency has been satisfactorily discussed and validated on different data-sets.

3. Briefly including sample use-cases of resulting structure would be useful to motivate future research directions.

4. Minor: Text in Fig. 1 and Fig. 2 is rather small, and quite difficult to read.

Author Response

Thanks for the comments.

4. Minor: Text in Fig. 1 and Fig. 2 is rather small, and quite difficult to read.

I mentioned that in the text that is remaining parameters, because i added 6 images in one table is like that but I changed the formatting to looks better.

Reviewer 4 Report

Authors presented a new approach “Optimizing Model Efficiency: Combining Knowledge Distillation and Network Pruning for Improved Generalization”. But paper needs some major modifications.

1.      The abstract and conclusion extensively need to be improved. The abstract must be a concise yet comprehensive reflection of what is in your paper. Please modify the abstract according to “motivation, description, results and conclusion” parts. I suggest extending the conclusions section to focus on the results you get, the method you propose, and their significance.

2.      What is the motivation of the proposed method? The details of motivation and innovations are important for potential readers and journals. Please add this detailed description in the last paragraph in Section 1. Please modify the paragraph according to "For this paper, the main contributions are as follows: (1) ......" to Section I. Please give the details of motivations. In Section 1, I suggest the authors can amend contributions of manuscript in the last of Section 1.

3.      I suggest in Section 2 (Related Work) please revise the content according to the development of timeline. Provide a critical review of the previous papers in the area and explain the inadequacies of previous approaches.

4.      The description of manuscript is very important for potential readers and other researchers. I encourage the authors to have their manuscript proof-edited by a native English speaker to enhance the level of paper presentation. There are some occasional grammatical problems within the text. It may need the attention of someone fluent in English language to enhance the readability.

5.      Please give the details of the proposed method for the proposed model. I suggest the authors amend the calculations of the proposed method and the details are important.

6.      However, the manuscript, in its present form, contains several weaknesses. Adequate revisions to the following points should be undertaken to justify recommendation for publication.

7.      In the conclusion section, the limitations of this study and suggested improvements to the presented work should be highlighted.

8.      Include latest references from 2023 focusing on Optimization and Network Pruning.

9.      Please cross check the scale of all figures in the manuscript to make them clear and more presentable.

10.   Please check all parameters in the manuscript and amend some related description of primary parameters. In section III, please write the proposed algorithm in a proper algorithm/pseudocode format. Otherwise, it is very hard to follow the flow of the study.

Moderate English language editing required.

Author Response

Hello,

Many thanks for the comments.

I changed whole the paper, abstract, introduction, conclusion and added more references and explained the gap between related works also added a figure for my proposed work , add two more figures for the discussion parts for understanding better.

for the figures they are 500 dpi, but because i put 6 images in one table they are small, I have changed them.

Round 2

Reviewer 1 Report

Congratulations to the authors for their effort in improving their paper. I find it suitable for publication in BDCC

Reviewer 2 Report

The authors have carefully revised the manuscript, I think it could be accepted as it is.

Reviewer 4 Report

Changes have been made as suggested.